Cell-free supernatants from cultures of lactic acid bacteria isolated from fermented grape as biocontrol against Salmonella Typhi and Salmonella Typhimurium virulence via autoinducer-2 and biofilm interference

Pelyuntha Wattana 1
Chaiyasut Chaiyavat 1
Kantachote Duangporn 2
Sirilun Sasithorn sasithorn.s@cmu.ac.th ssirilun@gmail.com 1
1 Innovation Center for Holistic Health, Nutraceuticals and Cosmeceuticals, Faculty of Pharmacy, Chiang Mai University , Chiang Mai , Thailand
2 Department of Microbiology, Faculty of Science, Prince of Songkla University , Hat Yai , Thailand
Sotelo-Mundo Rogerio
Electronic publication date: 2019 Aug 26
Publication date: 2019
Volume: 7
Electronic Location ID: e7555
Received 2019 Mar 29; Accepted 2019 Jul 25
Copyright: ©2019 Pelyuntha et al.
Copyright year: 2019
Copyright holder: Pelyuntha et al.
License: This is an open access article distributed under the terms of the Creative Commons Attribution License, which permits unrestricted use, distribution, reproduction and adaptation in any medium and for any purpose provided that it is properly attributed. For attribution, the original author(s), title, publication source (PeerJ) and either DOI or URL of the article must be cited.
License URL: https://creativecommons.org/licenses/by/4.0/

Keywords: Beneficial bacteria, Autoinducer-2, Biofilm, Salmonella, Lactic acid bacteria, Quorum sensing system

Funding: 50th Anniversary Chiang Mai University-Ph.D. scholarship Ph.D.010/2556 National Research Council of Thailand (NRCT) 2560A10402021 This study was financially supported by 50th Anniversary Chiang Mai University-Ph.D. scholarship (grant number: Ph.D.010/2556) and National Research Council of Thailand (NRCT) (grant number: 2560A10402021). The funders had no role in study design, data collection and analysis, decision to publish, or preparation of the manuscript.

==============================
Background

Salmonella Typhi and Salmonella Typhimurium are the causative pathogens of salmonellosis, and they are mostly found in animal source foods (ASF). The inappropriate use of antibiotics enhances the possibility for the emergence of antibiotic resistance in pathogens and antibiotic residue in ASF. One promising alternative to antibiotics in animal farming is the use of lactic acid bacteria (LAB).

Methods

The present study was carried out the cells and/or the cell-free culture supernatants (CFCS) from beneficial LAB against S. Typhi and S. Typhimurium. The antibacterial mechanisms of LAB-CFCS as biocontrol agents against both Salmonella serovars were investigated through the analysis of anti-salmonella growth activity, biofilm inhibition and quorum quenching activity.

Results

Among 146 LAB strains isolated from 110 fermented food samples, the 2 strong inhibitory effect strains (WM33 and WM36) from fermented grapes against both Salmonella serovars were selected. Out of the selected strains, WM36 was the most effective inhibitor, which indicated S. Typhi by showing 95.68% biofilm inhibition at 20% biofilm inhibition concentration (BIC) and reduced 99.84% of AI-2 signaling interference. The WM33 was the best to control S. Typhimurium by producing 66.46% biofilm inhibition at only 15% BIC and 99.99% AI-2 signaling a reduction. The 16S rDNA was amplified by a polymerase chain reaction (PCR). The selected isolates were identified as Weissella viridescens WM33 and Weissella confusa WM36 based on nucleotide homology and phylogenetic analysis.

Conclusion

The metabolic extracts from Weissella spp. inhibit Salmonella serovars with the potential to be used as biocontrol agents to improve microbiological safety in the production of ASF.

Introduction

Foodborne diseases are a serious cause of human illness and mortality. Salmonella spp. are major foodborne pathogens that cause a high rate of disease in humans and animals worldwide (Havelaar et al., 2015). Animal source foods (ASF) are rich sources of these pathogens; hence, they play a key role in spreading them. In the production of food products, Salmonella spp. could be found during the pre-harvest processes and subsequent stages of the production-to-consumption chain (Alum, Chukwu & Ahudie, 2016). In Southeast Asia, this is a public health concern, which has become more severe, because of the acceleration of Salmonella resistance to the common antibiotics used (Bhatia & Narain, 2010; Chuanchuen et al., 2010; Ellerbroek et al., 2010). Salmonella bacteria have been implicated as the causative agents in a spectrum of diseases, including enteric or typhoid fever (primarily Salmonella Typhi and S. Paratyphi), bacteremia, endovascular infections, enterocolitis (typically S. Typhimurium, S. Enteritidis, and S. Heidelberg) and asymptomatic carriers (Pui et al., 2011). Most frequently, humans become infected through the consumption of contaminated foods and water, especially livestock and their products. Unhygienic kitchens, restaurants and food industries can also lead to significant outbreaks (Eng et al., 2015).

Salmonella spp. have evolved mechanisms to enter host cells and involve intracellular rearrangement of host actin cytoskeleton, leading to food-poisoning related symptoms such as diarrhea, fever, abdominal cramp, abnormal stomach and vomiting (Pui et al., 2011). In severe cases, the patient becomes dangerously dehydrated, sepsis and carrier state may also develop (López et al., 2012).

One such mechanism is the quorum sensing (QS) system or cell-to-cell communication. This mechanism involves bacterial synthesis, secretion, and detection of small diffusible signal molecules known as autoinducers (AI) (Miller & Bassler, 2001). When the signal molecules reach critical threshold concentrations, AI can be detected and affect QS signaling cascade, which results in a change of the target gene expression, especially virulence genes (Rutherford & Bassler, 2012). Three types of the AI signaling molecules are frequently used by gram-negative bacteria as AI-1/LuxIR system, AI-2/LuxS system, and AI-3 QS system (Parker & Sperandio, 2009). The AI-1/LuxIR system has focused on LuxI, which synthesizes AI-1 or N-acyl homoserine lactone (AHLs), and LuxR, a transcriptional regulator responsible for activating of gene expression. The AI-1/LuxIR signaling pathway mediates intraspecies-specific communication (Ng & Bassler, 2009). Salmonella does not produce AHLs, but it can recognize AHLs from other bacterial species by SdiA, a LuxR homolog. SdiA-based QS system in Salmonella, which regulates several virulence genes located in virulence plasmid such as rck, which supports Salmonella in the evasion of human immune response (Ahmer et al., 1998; Parker & Sperandio, 2009).

The LuxS/AI-2 system has been discovered in both Gram-positive and Gram-negative bacteria and is well-documented as the universal QS system. LuxS-dependent AI-2 is produced by Salmonella during exponential growth and is released into the environment via a membrane transporter protein. Extracellular AI-2 can bind autoinducer binding protein LsrB and is transported into bacterial cells via Lsr transporter apparatus encoded on lsr operon (Pui et al., 2011). Salmonella bacteria use the AI-2/LuxS system to control the expression of virulence genes within SPI-1, which is responsible for Salmonella invasion (Choi, Shin & Ryu, 2007). In addition, AI-3 has also been discovered in a number of commensal bacteria, especially in Enterobacteriaceae; however, the synthetic pathway is still unclear (Parker & Sperandio, 2009). AI-3 is recognized through the two-component regulators comprised of histidine sensor kinases QseC and response regulator QseB, and then affected signaling cascade. This system activates the expression of genes responsible for flagella biosynthesis and bacterial motility (Parker & Sperandio, 2009). AI-3 regulators are also associated with the recognition of host epinephrine, norepinephrine and catecholamines, which induce SPI-2 gene expression to support Salmonella survival in macrophage, as well as facilitate the expression of genes encoded on SPI-1 and SPI-3 (Sperandio et al., 2003; Bearson & Bearson, 2008; Moreira, Weinshenker & Sperandio, 2010; Gart et al., 2016).

It has been reported that a variety of bacterial phenotypes and virulence factors, such as antibiotic production, sporulation, conjugation, motility, competence, bioluminescence, and biofilm formation are regulated in response to signaling molecules of QS systems (Rutherford & Bassler, 2012). In addition, several serovars of Salmonella are capable of attaching and forming biofilm on different surfaces (Steenackers et al., 2012). Biofilm formation is an important virulence factor and is well-known as the protective materials produced by bacteria to protect themselves against environmental stresses, antibiotics and host immune responses (Donlan, 2000; Jamal et al., 2015).

According to the microbiological food safety policy, many countries are concerned about Salmonella contamination in ASF production. All food industries emphasize microbiological food safety to control the level of pathogenic contamination in animal food production, and to decrease the risk factors that cause foodborne diseases and are associated with human illness, mortality, morbidity, and economic losses (Sousa, 2008; Hussain & Dawson, 2013). Food hygiene practices can reduce the spread of contamination and guarantee the safety of food (Caselli, 2017; Alum, Chukwu & Ahudie, 2016).

Furthermore, a range of management strategies, such as antibiotics, disinfectants, antimicrobial peptides, specific antibodies, vaccination, bacteriophage, and microflora, has been developed and scrutinized for their ability to decrease the risk factors that are related to contamination (Oh & Park, 2017). LAB strains are beneficial microflora and are between the most appropriate choice for application as living biocontrol bacteria for Salmonella management.

LAB strains are used in fermented foods as a way to extend shelf life and to improve the nutritional value and sensory characteristics. Some LAB strains are defined as probiotics and may promote health of the host’s gut (Quinto et al., 2014; Pandey, Naik & Vakil, 2015). They display diverse antagonistic mechanisms to defend against pathogenic bacteria. Possible mechanisms include nutrient competition; competition for adhesion sites; converting sugar to organic acids, which reduce the pH value; and forming a biological barrier to protect the host’s epithelial cells (Both, Abrahám & Lányi, 2011). In addition, LAB can produce a variety of antimicrobial substances, also known as natural preservatives, such as organic acids, hydrogen peroxide, antimicrobial peptides, and bacteriocins. These substances play an important role in inhibiting the growth of spoilage and pathogenic bacteria in fermented foods (Reis et al., 2012).

Strategies that focus on LAB and their metabolites to perturb AI-2 signaling activity and biofilm formation of Salmonella, have been suggested as suitable strategies for controlling Salmonella, and can attenuate target bacterial virulence factors.

The aim of this study is to evaluate the antagonistic properties of LAB and their metabolic substances (CFCS) against Salmonella (S. Typhi and S. Typhimurium) growth and their virulence factors via AI-2/LuxS system and biofilm interference.

Materials & Methods

Bacterial strains used

Salmonella Typhi DMST 22842 and Salmonella Typhimurium TISTR 1469 were used in present study. Both Salmonella indicators, which are the major serovars of Salmonella enterica that cause foodborne illnesses in humans, were streaked on to Tryptic Soy agar (TSA) and a single colony was grown in Tryptic Soy broth (TSB) (Merck, Darmstadt Germany) for 24 h at 37 °C. Vibrio harveyi BB170 (ATCC BAA-1117) acts as a reporter strain (AI-2 sensor positive), which exhibits the bioluminescent activity after the recognition of AI-2 molecules, and V. harveyi BB152 (AI-2 producer) served as the positive control. They were cultured in Zobell Marine broth 2216 for 18 h at 30 °C with a shaking incubator. All bacterial strains used in this study were obtained from the Department of Pharmaceutical Sciences, Faculty of Pharmacy, Chiang Mai University, Thailand.

Isolation of LAB from fermented foods

A total of 110 samples of various fermented foods including 10 local fermented fish products (five samples of Pla-ra and five samples of Plaa-som), 20 traditional fermented meat products (10 samples of Nham (fermented ground pork sausage), five samples of Sai-Krork-Prew (fermented pork sausage), and five samples of Mum (fermented pork meat with liver and spleen)), 20 fermented soybean products (five samples of fermented pickled soybeans, five samples of Tao-hoo-yee (fermented bean curd), and 10 samples of Tooa-nao (Thai fermented soybean)), 46 fermented vegetables (15 samples of pickled cabbages, 15 samples of pickled mustard greens, 10 samples of pickled cucumbers, two samples of dried salted Chinese radish, one sample of fermented bamboo shoot, and three samples of pickled garlic bulbs), and 14 fermented fruits (eight samples of pickled mango, four samples of pickled grape and two samples of pickled tamarind) were collected randomly from the local markets in Chiang Mai province, Thailand. All samples were kept in sterile containers, transported to the laboratory and maintained at 4 °C until analysis. Then, 25 g of each sample was homogenized in 225 mL of phosphate buffer saline (PBS), pH was 7.2 ± 0.2. All LAB strains were isolated and purified on de Man Rogosa Sharpe (MRS) agar (Difco, Detroit, Michigan, USA) with 0.005% (w/v) bromocresol purple. All plates were incubated at 37 °C for 24–48 h. LAB isolates were initially characterized by Gram’s staining reaction and catalase test (Ben Slama et al., 2013). All Gram-positive and catalase-negative isolates were maintained in MRS broth with 15% (v/v) glycerol at −20 °C. Before being used in the experiment, these stocks were sub-cultured twice in the MRS broth for obtaining an active culture.

Agar spot test

The antagonistic activities of the isolated LAB against the growth of both target organisms were determined by the agar spot test following the procedures of Djadouni & Kihal (2012) with slight modification. Isolated LAB were cultivated in MRS broth at 37 °C for 24 h of incubation; 3 µL of each culture broth was spotted onto the surface of TSA on which was poured a suspension of either S. Typhi or S. Typhimurium at a final concentration of 105 CFU/mL. All plates were incubated at 37 °C for 48 h. The inhibition or halo zone around the dropped colonies was defined as the level of antimicrobial activity against the growth of indicator strains.

Agar well diffusion test

The inhibitory activities of the LAB supernatants to Salmonella were determined by agar well diffusion test as described by Gaamouche et al. (2014). Isolated LAB were propagated in MRS broth at 37 °C for 24 h. Cell-free culture supernatant (LAB-CFCS) was collected and filtered through a sterile syringe micro-filter of 0.22 µM pore size. TSA soft agar (1% agar) was mixed with a final approximate concentration of 105 CFU/mL of an indicator strain. A 20 ml of soft agar mixture was poured into a sterile petri-dish containing 12 stainless-steel carriers (5 mm in diameter). After setting of the agar medium, wells were formed by pulling out the carriers; 50 µL of CFCS was filled into the TSA agar wells, and then the plates were incubated for 48 h at 37 °C. The inhibitory spectrum of LAB-CFCS around the wells was measured with Vernier caliper and recorded.

Minimum inhibitory (MIC) and minimum bactericidal (MBC) concentrations

The MIC value of individual LAB-CFCS against indicator strains was assessed in 96-well plates as described by Ben Taheur et al. (2016) with slight modification. CFCS were serially diluted in TSB for indicator strains to yield a final concentration ranging from 5% to 90% (v/v) of CFCS, and then 10 µL of each indicator was added to each well. The total volume of each well was 200 µl, and the final concentration of indicator strain at 105 CFU/ml. MIC value was defined as the lowest concentration of LAB-CFCS by showing no turbidity. MBC can be determined by sub-culturing 5 µL of each sample from a MIC micro-dilution test well, yielding a negative microbial growth after incubation on the surface of TSA plates to determine the surviving bacterial cells after 24 h at 37 °C of incubation. The bactericidal endpoint (MBC) is subjectively defined as the lowest concentration at which 99.9% of the final inoculum is killed.

Anti-biofilm activity of Salmonella by LAB-CFCS

The action of LAB-CFCS against biofilm formation of both pathogens was tested on 24-well micro-plates cell culture (NUNCLON™ delta Surface #143982, Nunc, Denmark). LAB-CFCS were serially diluted at concentrations of 5–40% (v/v) in Luria-Bertani (LB) broth containing the Salmonella suspension at 106 CFU/mL. The plates were incubated at 37  °C for 48 h to allow cell attachment and biofilm development. After incubation, BIC was determined as the lowest concentration that produces visible disruption in biofilm formation (Thenmozhi et al., 2009). For quantitative analysis, wells containing biofilm at various percentage of BIC were investigated by crystal violet staining assay using spectrometric quantitation. The remaining planktonic cells in the medium were aspirated, and wells were washed three times with normal saline solution (NSS, 0.85% NaCl). Then, 0.1% crystal violet solution in water was added for 30 min, washed three times with NSS and allowed to dry. Finally, 1 mL of 95% ethanol was added to destain the well; 200 µL of solution was transferred to the new 96-well micro-plate and quantified in SoftMax® Pro7 by SpectraMax M3 micro-plate reader (Molecular Devices, USA) at 545 nm. The results were expressed as the percentage of biofilm inhibition:

% biofilm inhibition = [(ODcontrol–ODBIC)/ODcontrol] × 100 (Ben Slama et al., 2013).

Detection of AI-2 activity in S. Typhi and S. Typhimurium

The AI-2 activity in both Salmonella indicators was investigated, and V. harveyi BB152 served as control. In this experiment, V. harveyi BB170 was used as the reporter strain to verify AI-2 signaling activity. V. harveyi BB170 exhibits bioluminescence in the presence of exogenous AI-2 molecules. S. Typhi and S. Typhimurium were grown in 5 mL TSB at 37 °C for 18 h; the culture supernatants were collected by centrifugation and filtered through a 0.22 µm syringe filter, and the AI-2 bioluminescence assay was performed. A 16 h growth of V. harveyi BB170 was freshly diluted in AB medium (1:5000); 90 µL of the diluted AB medium was dispensed into 96-well luminescent micro-plates (Nunc™ F96 MicroWell™ #236108. Nunc, Denmark). A quantity of 10 µL of each Salmonella supernatant was added into the wells, and the bioluminescence was measured as a relative light unit (RLU) at 30 min-interval for 6 h with SoftMax® Pro7 by SpectraMax M3 microplate luminometer (Molecular Devices, San Jose, CA, USA). Wells containing the supernatant of V. harveyi BB152 and fresh AB medium served as positive and negative controls respectively. The percentage of AI-2 signaling activity was calculated with RLU at 6 h with the formula as follow:

% AI-2 signaling activity = [RLUSalmonella/RLUpositive] × 100 (Sivakumar, Jesudhasan & Pillai, 2011).

Interference test of AI-2 signaling in Salmonella by LAB-CFCS

To interfere with AI-2 signaling activity in Salmonella indicators, four selected LAB-CFCS were used in this study. As described in the previous test, the diluted culture BB170 in AB medium was dispensed into 96-well luminescent micro-plates; 5 µL of Salmonella supernatant and 5 µL of LAB-CFCS were added into the wells. The bioluminescence activity of the mixture was measured using a luminometer. In addition, the positive control was a mixture of 5 µL of Salmonella supernatant and 5 µL of AB medium, while the negative control was 10 µL of AB medium. The results were calculated and expressed as the percentage reduction in AI-2 activity using this formula:

%AI-2 signaling interference = [(RLUpositive–RLULAB-CFCS∕)RLUpositive] × 100 (Widmer et al., 2007; Soni et al., 2008).

Identification of selected LAB strains

Two LAB strains were firstly tested for their biochemical and physiological properties following the methods as described by Liu et al. (2014). Each strain was grown in MRS broth at 37 °C for 24 h, transferred into five mL MRS broth and incubated at 15, 37 and 45 °C for 24–48 h. The 6.5 and 18% (w/v) NaCl tolerance test of LAB was also performed. A sugar fermentation test was carried out in 96-well plates; the modified MRS broth containing bromocresol purple (0.0025%), with glucose omitted, were mixed with 10% (w/v) sterile sugar solution (9:1), including galactose, lactose, maltose, mannitol, mannose, raffinose, sucrose, arabinose, sorbitol, and xylose, to obtain 1% sugar concentration. A 180 µL of each sugar solution was dispensed into wells, and then 20 µL of strain WM33 or WM36 was inoculated and incubated at 37 °C for 48 h. A change of colour was observed and interpreted as LAB having the ability to assimilate those sugars as a carbon source.

For molecular identification, genomic DNA of each potent LAB was extracted and purified using Nucleospin® DNA kit according to the manufacturer’s instructions. The full-length of 16s rRNA gene (∼1,500 base pairs) was sequenced on both strands of PCR-amplified fragments, and was performed using the dideoxy chain termination method by the commercial service of Macrogen Inc. (Seoul, Korea). DNA sequences were edited, and consensus sequences were obtained using the Bioedit software package. Final sequences were then aligned using CLUSTAL for each of the sequences (Tilahun et al., 2018). The sequences of both potent LAB isolates were compared to those in the Genbank nucleotide database (http://www.ncbi.nih.gov/) with the Basic Local Alignment Search Tool for nucleotide sequences (blastn) of the National Center for Biotechnology Information (NCBI, USA). Phylogenetic tree construction was performed using the Neighbor-Joining method based on the Kimura 2-parameter model with MEGA-X (Kumar et al., 2018).

Statistical analysis

Statistical analysis was performed using the SPSS version 17.0 of Windows (SPSS Inc, Chicago, IL, USA). Values were expressed as mean ± standard deviation of triplicate. A statistical comparison was performed by the one-way analysis of variance (ANOVA), followed by the Tukey’s HSD test. The results were considered statistically significant when the p-values were less than 0.05.

Results

Antibacterial activity of isolated LAB

A total of 19 LAB from 146 isolates showed antibacterial activity against S. Typhi, while only seven isolates showed antibacterial activity against S. Typhimurium using the agar spot test. An example of the inhibitory zone is shown in Fig. S1. All LAB isolates that inhibited the growth of S. Typhimurium are subsets of those that inhibited S. Typhi activity, including WM13, WM19, WM21, WM24, WM33, WM34, and WM36. For secondary screening with agar well diffusion assay (Tables 1; S1), the ability of LAB-CFCS to inhibit the growth of Salmonella indicators was investigated. Among them, 16 of the 19 isolates still kept their inhibitory activity against S. Typhi, while four of the seven isolates kept their activity against S. Typhimurium. The average diameter of the inhibitory zone ranged from 3-9 mm in size (Table 1; Fig. S2).

Table 1 Inhibitory spectrum of cell-free culture supernatant (CFCS) from LAB against Salmonella indicators by agar well diffusion test.

Each value is provided as the mean ± standard deviation of triplicate, and those connected by the different letters in the same column are significantly different (p < 0.05). (no activity, no inhibition zone; -, not performed due to negative effect on S. Typhimurium growth). The asterisk (*) indicates the strains that were selected for further studies.

LAB-CFCS	Zone of inhibition (mm)	
	S.Typhi	S.Typhimurium	
WM1	4.83 ± 0.29c	–	
WM2	3.50 ± 0.00ab	–	
WM3	3.50 ± 0.70ab	–	
WM5	4.83 ± 0.76c	–	
WM6	3.83 ± 0.58abc	–	
WM8	3.35 ± 0.39a	–	
WM11	3.50 ± 0.50ab	–	
WM12	3.83 ± 0.29abc	–	
WM13	no activity	no activity	
WM19 ∗	3.75 ± 0.25abc	6.25 ± 0.18b	
WM21	no activity	no activity	
WM24	no activity	no activity	
WM33 ∗	3.50 ± 0.00ab	4.25 ± 0.22a	
WM34 ∗	3.00 ± 0.00a	5.25 ± 0.15ab	
WM36 ∗	3.17 ± 0.29a	6.25 ± 0.75b	
PR1	6.17 ± 0.29d	–	
PR2	6.67 ± 0.29d	–	
PR14	4.67 ± 0.29bc	–	
FC14	6.00 ± 0.00d	–	
MRS broth	no activity	no activity	
Ampicillin	7.75 ± 0.75e	9.00 ± 1.14b	

Table 2 shows the results of MIC and MBC values; 16 selected LAB-CFCS displayed MIC values that ranged from 10% to 60% for S. Typhi and 20% to 40% for S. Typhimurium. In addition, the MBC values ranged from 20% to 80% for S. Typhi and 30% to 40% for S. Typhimurium.

Table 2 Minimum inhibition concentration (MIC) and minimum bactericidal concentration (MBC) values of cell-free culture supernatant from LAB against Salmonella indicators.

All values are provided as mean ± standard deviation of triplicate. (-, not performed due to negative effect on S. Typhimurium growth. The asterisk (*) indicates the strains that were selected for further studies.

LAB-CFCS	S.Typhi	S.Typhimurium	
	MIC (%)	MBC (%)	MIC (%)	MBC (%)	
WM1	40.00 ± 0.00	50.00 ± 0.00	–	–	
WM2	60.00 ± 0.00	60.00 ± 0.00	–	–	
WM3	20.00 ± 0.00	40.00 ± 0.00	–	–	
WM5	20.00 ± 0.00	40.00 ± 0.00	–	–	
WM6	25.00 ± 0.00	40.00 ± 0.00	–	–	
WM8	40.00 ± 0.00	40.00 ± 0.00	–	–	
WM11	20.00 ± 0.00	40.00 ± 0.00	–	–	
WM12	60.00 ± 0.00	80.00 ± 0.00	–	–	
WM19 ∗	20.00 ± 0.00	40.00 ± 0.00	20.00 ± 0.00	30.00 ± 0.00	
WM33 ∗	20.00 ± 0.00	40.00 ± 0.00	20.00 ± 0.00	30.00 ± 0.00	
WM34 ∗	20.00 ± 0.00	40.00 ± 0.00	20.00 ± 0.00	30.00 ± 0.00	
WM36 ∗	40.00 ± 0.00	40.00 ± 0.00	40.00 ± 0.00	40.00 ± 0.00	
PR1	30.00 ± 0.00	40.00 ± 0.00	–	–	
PR2	30.00 ± 0.00	30.00 ± 0.00	–	–	
PR14	10.00 ± 0.00	20.00 ± 0.00	–	–	
FC14	20.00 ± 0.00	20.00 ± 0.00	–	–	
MRS broth	no activity	no activity	no activity	no activity	

Overall the results of four LAB strains (WM19, WM33, WM34, and WM36) with their metabolites showed a strong inhibition against both Salmonella serovars, which were selected for further study.

Anti-biofilm activity by LAB-CFCS

Four LAB-CFCS had the ability to act as a potential alternative strategy for biofilm inhibition in both Salmonella indicators (Tables 3; S2; Figs. S3; S4). Individual LAB-CFCS showed different BIC values ranging from 20% to 30% for anti-biofilm activity against S. Typhi with 95% to 96% to inhibit biofilm formation. Based on the percentage of biofilm inhibition, anti-biofilm of S. Typhi by strains WM34 and WM36 were significantly higher (p < 0.05) than that found by WM33 and WM19. However, strain WM36 was more effective than strain WM34 with a lower of BIC percentage. For S. Typhimurium, LAB-CFCS exhibited BIC values of 15% to 20% with significantly different percentages of biofilm inhibition, ranging from 46% to 75%, and the inhibition was in the order of strains WM36 >WM33 >WM34 >WM19. It should be noted that among them, strain WM33 used only 15% BIC for 66% inhibition, while strain WM36 used 20% BIC for 75% inhibition.

Table 3 Biofilm inhibition concentration (BIC) values and % biofilm inhibition at BIC of LAB-CFCS against biofilm production of Salmonella indicators.

The percentages of biofilm inhibition are provided as the mean ± standard deviation of triplicate, and those connected by the different letters in the same column are significantly different (p < 0.05). The asterisk (*) indicates the strains that were selected for further studies.

LAB-CFCS	S.Typhi	S.Typhimurium	
	BIC (%)	% biofilm inhibition	BIC (%)	% biofilm inhibition	
WM19	20	94.85 ± 0.25a	15	45.92 ± 0.77a	
WM33 ∗	20	94.98 ± 0.04a	15	66.46 ± 0.19c	
WM34	30	96.09 ± 0.06b	20	52.74 ± 0.15b	
WM36 ∗	20	95.68 ± 0.27b	20	74.83 ± 0.15d	

Interference test of AI-2 signaling in Salmonella by LAB-CFCS

The AI-2 signaling activity in S. Typhi and S. Typhimurium supernatants was determined. The results show that S. Typhi and S. Typhimurium produce significant amounts of AI-2 signaling activity as 47.49 ±3.23% and 52.17 ±1.33% respectively, compared with the positive control (V. harveyi BB152), which is normalized as 100% of activity (Table S3 ).

The interference of AI-2 signaling molecules in Salmonella may affect QS-associated behaviors and/or biofilm formation. Tables 4 and S4 show the percentage interference of AI-2 activities of S. Typhi and S. Typhimurium in the presence of only 5% LAB-CFCS. All LAB-CFCS exhibited very high inhibition, about 99%, of both Salmonella serovar and did not interfere with the growth of reporter V. harveyi BB170 (Table S5). Our results suggest that the metabolites in LAB-CFCS may also exert quorum quenching action.

Table 4 The percentage of AI-2 signaling interference against Salmonella by LAB-CFCS.

All values are provided as mean ± standard deviation of triplicate, and those connected by the different letters in the same column are significantly different (p < 0.05). The asterisk (*) indicates the strains that were selected for further studies.

LAB-CFCS	% AI-2 signaling interference	
	S.Typhi	S.Typhimurium	
WM19	99.58 ± 0.05b	99.98 ± 0.01bc	
WM33 ∗	99.41 ± 0.08ab	99.99 ± 0.00c	
WM34	99.19 ± 0.07a	99.98 ± 0.00b	
WM36 ∗	99.84 ± 0.12c	99.97 ± 0.00a	

Identification of selected LAB strains

Based on the anti-biofilm activity test, strains WM34 and WM36 showed the highest percentage of biofilm inhibition (roughly 96%) against S. Typhi; however, the latter strain used only 20% BIC compared with 30% BIC of the former strain (Table 3). Moreover, WM36 also showed the highest percentage of biofilm inhibition at 75% with 20% BIC against S. Typhimurium. Nevertheless, strain WM33 showed 66% biofilm inhibition against S. Typhimurium at only 15% BIC. In the case of AI-2 signaling interference, WM36 still kept the highest AI-2 signaling interference against S. Typhi and WM33 showed the highest AI-2 signaling interference in S. Typhimurium (Table 4). Therefore, WM33 and WM36 were chosen as potent LAB strains and used for bacterial identification.

On the basis of morphological, physiological and biochemical characteristics, strains WM33 and WM36 presented Gram-positive and catalase-negative behavior. WM33 showed rod-shaped, while WM36 showed coccobacilli-shaped morphology and they produced CO2, which are classified as heterofermentative LAB. In addition, both LAB strains also showed different patterns of carbohydrate fermentation; growth at different temperatures; and NaCl tolerance; as detailed in Table 5 and Dataset S1.

Table 5 The fundamental characterization of LAB strains.

Characteristics	WM33	WM36	
Gram’s strain	+	+	
Shape	R	CB	
Catalase	−	−	
Gas production from glucose	+	+	
Carbohydrate fermentation	
Maltose	+	+	
Mannitol	−	−	
Lactose	−	−	
Xylose	−	+	
Sucrose	−	+	
Sorbitol	−	−	
Arabinose	−	−	
Raffinose	−	−	
Mannose	−	+	
Galactose	−	+	
Growth at different temperature (°C)	
15	−	+	
45	−	+	
Salt tolerance (% w/v)	
6.5	−	−	
18	−	−	
Notes.

R rod

CB coccobacilli

+ present/growth

- absence/no growth

The results obtained from the sequencing analysis of 16s rRNA genes, a phylogenetic tree was constructed and was shown in the Fig. 1 and Dataset S1. The WM33 isolate was identified as Weissella viridescens with 100% similarity (NCBI accession number: MK680135.1) and WM36 isolate showed 100% similarity to Weissella confusa (NCBI accession number: MK680136.1) in the GenBank database. The original habitat of both isolates was from fermented grape, but different samples. Moreover, W. viridescens WM33 and W. confusa WM36 are permanently deposited in the Thailand Bioresource Research Center (TBRC), Pathum Thani, Thailand with the accession numbers TBRC11085 and TBRC11086, respectively.

Figure 1 Neighbor-joining phylogenetic tree based on 16s rRNA gene sequence analysis of WM33 and WM36 (1,249 bp aligned).

Bootstrap values >50% based on 1,000 replicates are shown at branch nodes Bar 0.05 substitutions per nucleotide position.

Discussion

Outbreaks of foodborne diseases involving Salmonella are serious problems worldwide, leading to significant economic and health issues. Although there are a number of alternative approaches developed to inhibit the growth of Salmonella in ASF production, these approaches have led to an increasing number of issues in food industries. For example, antibiotic use in animal agriculture leads to an increase of antibiotic-resistant bacteria and subsequent resistance in humans; the cost of vaccinations and treatments are high; and the medical administration programs are complex. Therefore, LAB and their metabolites in CFCS are a viable alternative to deal with these problems. In addition, they are safer and are more easily administered than others.

The results of the agar spot and agar well diffusion tests indicate that the inhibitory activity against both Salmonella indicators was mostly due to LAB-CFCS. This suggests that LAB metabolites in LAB-CFCS play a major role in anti-salmonella activity.

To support our results, we point to the antimicrobial activity of LAB and CFCS against Salmonella sp. that has been reported previously. The research of Casey et al. (2004) reveal that 26 LAB isolates exhibited great anti-salmonella activity and the inhibition zone ranged from 4 to 9 mm. Tatsadjieu et al. (2009) report that their LAB isolate, named LF2, was active against S. enterica with a larger inhibition zone of more than 25 mm, and they conclude that the inhibitory activity is due to the biological activity of bacteriocin. Moreover, Li, Gu & Zhou (2016) reveal that the CFCS of Lactobacillus plantarum LZ206 show antibacterial activity (ranging from 20–25 mm) against S. enterica due to its bacteriocin. Our LAB-CFCS exhibited lower activity as inhibition zones in a range of 3 to 7 mm against S. Typhi and S. Typhimurium (Table 1) compared with Tatsadjieu et al. (2009) and Li, Gu & Zhou (2016) studies. It is well recognized that anti-salmonella by LAB-CFCS depends on the virulence of the pathogenic strains tested, and also on the bioassay methods used. As agar well diffusion was used, it would be possible that metabolites in our LAB-CFCS may have a low solubility in agar. This hypothesis was confirmed in the next experiments (MIC and MBC tests) for bioassay in broth.

In order to understand the antibacterial efficiency of LAB-CFCS, MIC and MBC values of four LAB-CFCS were determined. Lactobacillus fermentum showed MIC value with 30% CFCS and 50% CFCS to complete growth inhibition of S. Typhimurium. Lactobacillus salivarius showed MIC value with 20% CFCS and complete inhibition of S. Typhimurium growth at 40% (Afdora et al., 2010). These results are similar to those of Ben Taheur et al. (2016), who report that three LAB isolates including Pediococcus pentosaceus FB2, exerted antibacterial activity against S. Typhimurium, which displayed MIC and MBC values of 60% and 60% respectively. Similarly, the MIC and MBC values of P. pentosaceus FG1 were 40% and 70% respectively, while those of Lactobacillus brevis FF2 were 70% and 100%, respectively. Four LAB-CFCS in this present study showed the antibacterial susceptibility effects on both Salmonella indicators at 20–40% for MIC value and 30–40% for MBC value (Table 2). The results confirm that metabolites in our LAB-CFCS had a limited solubility in agar as previously discussed. This is due to their MIC and MBC values when testing in broth being more effective than in the above previous studies for inhibiting Salmonella spp.

Several studies have reported the application of LAB from various sources that possess anti-biofilm activity. Probiotic LAB strains have been investigated for their anti-biofilm activity against a wide range of biofilm-producing pathogens. For example, Lactobacillus rhamnosus GG produces lectin-like molecules that showed inhibitory activity against S. Typhimurium biofilm (Petrova et al., 2016). Crude bacteriocin from L. brevis DF01 could inhibit the biofilm formation of S. Typhimurium on stainless steel (Kim, Kim & Kang, 2019). Organic acids, namely lactic acid, acetic acid, and citric acid, are efficient inhibitor against Salmonella spp. These organic acids show the maximum biofilm inhibition ranged from 13% to 39% by decreasing the exopolysaccharide production, which is the main component of Salmonella biofilms (Amrutha, Sundar & Shetty, 2017). Normally, organic acids are the main metabolites that all LAB use for antimicrobial activity and lower the pH in fermented foods. It is suggested that in the presence of organic acids such as lactic acid and acetic acid, cells may become inactivated and lead to inhibition biofilm formation processes (Akbas, 2015).

Our research shows that our LAB-CFCS are promising strategies for Salmonella biofilm inhibition (95–96% for S. Typhi and 46–75% for S. Typhimurium as shown in Table 3). Salmonella can grow in a broad pH range of 4–9, with the optimum being 6.5–7.5 and do not survive in acidic environments (El Hussein et al., 2012). Our tested LAB-CFCS were very effective to inhibit biofilm formation of both Salmonella serovars.

Several reports have extensively investigated the AI-2 activity in Salmonella bacteria. In this present study both Salmonella serovars produced AI-2 signaling activity ranging from 47.5% to 52.2%. This concurs with Almasoud et al. (2016) who report that in their study, S. Typhimurium SD10 and SD11 produced 53.2% and 21.3% of AI-2 signaling activity, respectively. Interfering with QS mechanisms has been found to be a more effective way to fight bacterial infections. LuxS/AI-2 is the universal QS signaling system, which can be found in numerous bacterial species, and is involved in the production and perception/response to exogenous AI-2. The perturbation of AI-2 signaling can provide an advantageous therapeutic strategy and have emerged as potential targets for anti-infective therapy in various bacterial infections (Xavier & Bassler, 2005; Reuter, Steinbach & Helms, 2016).

The role of a variety of natural, synthetic and pure compounds as AI-2 inhibitors have been demonstrated, e.g., proteins, fatty acids, phytochemical extracts as well as organic acids. Poultry meat-derived fatty acids, including linoleic acid, oleic acid, palmitic acid, and stearic acid, can also inhibit AI-2 signaling ranging from 25% to 99% (Widmer et al., 2007). Acid food preservatives such as sodium propionate and sodium benzoate also reduced the AI-2 like activity by 75% to 99% (Lu, Hume & Pillai, 2004). In addition, natural organic acids are also considered as AI-2 inhibitors, including lactic acid and malic acid. These are effective in inhibiting AI-2 activities of S. Typhimurium and yielded a high inhibition of 80% (Almasoud et al., 2016). The primary metabolites secreted by LAB are usually an organic acid group. The mode of action of these molecules is thought to be pH dependent. Organic acids may affect the redox reductions of NADPH formation, which is particularly required as energy sources for bacterial bioluminescence in the reporter strain (Almasoud et al., 2016). Our LAB-CFCS yielded the greatest reduction of AI-2 signaling activity by 99% in both Salmonella indicators; thereby the putative LAB-CFCS are candidate quorum quenching agents.

Antimicrobial resistance in Salmonella can be associated with horizontal transference of antibiotic-resistant genes. Salmonella is resilient bacteria with a complex genomic system that enables the organism to react to different environmental conditions and antimicrobial agents (Andino & Hanning, 2015). Several mechanisms of Salmonella to develop resistance antimicrobial agents include production of enzymes that can degrade cell permeability to antibiotics; activation of antimicrobial efflux pumps; the production of enzymes to degrade the chemical structure of antimicrobial agents (Andino & Hanning, 2015)); and biofilm formation that serve to protect them from external adverse influences and enhance bacterial resistance to antibiotics and sanitizers (Donlan, 2000; Jamal et al., 2015). Furthermore, the virulence of S. Typhi and S. Typhimurium depends on the activity of signaling molecules, autoinducer 2 (AI-2) via the luxS synthase gene, which is used by some pathogens to coordinate the virulence gene expression with density of colonization (Choi, Shin & Ryu, 2007). In many research studies, nontyphoidal serovars such as S. Typhimurium has also been found to show high rates and severity of resistance to the traditional antimicrobials, and resistance to some antibiotics have been found to have emerged in several countries (Andino & Hanning, 2015; VT Nair, Venkitanarayanan & Kollanoor Johny, 2018). From our results, each LAB strains may show better performance in some aspects of the reported mechanisms (anti-QS and anti-biofilm formation) of LAB against two Salmonella serovars, and the underlying mechanisms of the preventive effects may be complex and intricately related. These might be reasons why LAB strains better against S. Typhi than S. Typhimurium.

Lactic acid from LAB is also known to function as a permeabilizer of the Gram-negative bacterial outer membrane allowing other compounds to perform synergistically with lactic acid (Alakomi et al., 2000). Lactic acid also specifically influences the expression of the Salmonella key virulence gene. The structure and amount of antimicrobial substances of LAB strains may account for the strain-specific properties, which might be the reasons why different strains of the same bacteria performed different antagonistic activity against bacterial pathogens of two different serovars of Salmonella.

From the preliminary identifications, two potent LAB isolates were identified as members of the genus Lactobacillus or Lactobacillus-like microorganisms (Liu et al., 2014). Molecular techniques correctly identified these strains as W. viridescens and W. confusa (Fig. 1). Prior better understanding, these Weissella strains were previously known as Lactobacillus viridescens and L. confusus, respectively (Fusco et al., 2015). This is why our results obtained from the biochemical and physiological tests initially identified WM33 and WM36 isolates as Lactobacillus or Lactobacillus-like microorganisms.

Several studies report that Weissella exerts antagonistic activity against foodborne pathogens. Some strains of Weissella are capable of producing antimicrobials, including weissellicin, or compete for pathogen adhesion sites (Abriouel et al., 2015; Fessard & Remize, 2017).

Therefore, the results obtained from all experiments in this present study confirm that W. viridescens WM33 and W. confusa WM36 isolated from the fermented foods are beneficial LAB, which can act as a source of anti-salmonella metabolites, particularly for preventing infection of Salmonella by reducing both AI-2 signaling and biofilm formations.

The results presented in this study demonstrate the ability of CFCS of some LAB isolates to inhibit growth, biofilm formation and virulence factors of Salmonella. However, the structural characterization; the amount of the active substances in CFCS; and the specificity of their antagonistic activity against Salmonella remain an important area for future research.

Conclusions

We successfully obtained two beneficial strains of LAB from fermented grape and their metabolites, which possess the ability to antagonize and interfere with the growth, biofilm formation, and QS regulation (via AI-2 signaling interference) of Salmonella pathogenic indicators. W. viridescens WM33 and W. confusa WM36 with their released metabolites have great potential to be used as biocontrol agents/biopreservatives for controlling Salmonella in ASF production to achieve microbiological safety of food.

Supplemental Information

Table S1 Raw data obtained from the agar well diffusion test for data analysis and preparation for Table 1

The mean and standard deviation of each treatment were calculated based on three replicates. In addition, the different significance in statistical analysis were also investigated at p-value less than 0.05.

Click here for additional data file.

Table S2 Raw data obtained from the biofilm inhibition assay for data analysis and preparation for Table 3

The mean and standard deviation of each treatment were calculated based on three replicates. In addition, the different significance in statistical analysis were also investigated at p-value less than 0.05.

Click here for additional data file.

Table S3 Raw data obtained from the AI-2 signaling activity in S. Typhi and S. Typhimurium

The mean and standard deviation of each treatment were calculated based on three replicate. In addition, the different significance in statistical analysis were also investigated at p-value less than 0.05.

Click here for additional data file.

Table S4 Raw data obtained from the interference of AI-2 signaling activity for data analysis and preparation for Table 4

The mean and standard deviation of each treatment were calculated based on three replicates. In addition, the different significance in statistical analysis were also investigated at p-value less than 0.05.

Click here for additional data file.

Table S5 The growth of reporter V. harveyi BB170 in the presence of 5% (v/v) CFCS

Bacterial growth curve was plotted by the normalized absorbance at 600 nm versus the incubation time.

Click here for additional data file.

Figure S1 The example of LAB colonies with inhibitory activity against Salmonella indicators via agar spot test

LAB colonies with inhibitory activity against Salmonella indicators are indicated by black arrow.

Click here for additional data file.

Figure S2 The example of LAB-CFCS with anti-salmonella growth using agar well diffusion test

Red rectangle indicates the agar wells containing LAB-CFCS with inhibitory activity against Salmonella indicators. Bubbles are made by gas production of Salmonella indicators.

Click here for additional data file.

Figure S3 The example of Salmonella biofilm development on 24-well microplate

Black arrow indicates the biofilm layer of Salmonella indicators.

Click here for additional data file.

Figure S4 Crystal violet staining assay for Salmonella biofilm inhibition

Microplate with biofilm after crystal violet staining.

Click here for additional data file.

Dataset S1 Raw data obtained from identification of selected LAB strains for Table 5 and Fig. 1

Raw data contained sugar fermentation test, Gram’s stain test, bacterial sequences and phylogenetic tree analysis.

Click here for additional data file.

We thank all the members in our academic group for helping us complete the experiments. The authors also acknowledge the Faculty of Pharmacy, Chiang Mai University, Chiang Mai, Thailand for kind assistance in allowing us to conduct the research work.

Additional Information and Declarations

Competing Interests

Author Contributions

DNA Deposition

Data Deposition

The authors declare there are no competing interests.

Wattana Pelyuntha conceived and designed the experiments, performed the experiments, analyzed the data, contributed reagents/materials/analysis tools, prepared figures and/or tables, authored or reviewed drafts of the paper, approved the final draft.

Chaiyavat Chaiyasut conceived and designed the experiments, authored or reviewed drafts of the paper, approved the final draft.

Duangporn Kantachote authored or reviewed drafts of the paper, approved the final draft.

Sasithorn Sirilun conceived and designed the experiments, performed the experiments, authored or reviewed drafts of the paper, approved the final draft.

The following information was supplied regarding the deposition of DNA sequences:

The sequences of two Weissella isolated described here are accessible via GenBank: MK680135.1 and MK680136.1.

The following information was supplied regarding data availability:

Bacterial isolates are deposited in Thailand Bioresource Research Center (TBRC), Pathum Thani, Thailand. Isolates are available at TBRC, accession number: TBRC11085 and TBRC11086.

http://www.tbrcnetwork.org/products.php?product_id=19290.

http://www.tbrcnetwork.org/products.php?product_id=19291.

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
