# Peer review of "Cell-free supernatants from cultures of lactic acid bacteria isolated from fermented grape as biocontrol against Salmonella Typhi and Salmonella Typhimurium virulence via autoinducer-2 and biofilm interference"

_PeerJ, doi:10.7717/peerj.7555_

## Round 0.1 · original submission · Minor Revisions

Please present a rebuttal letter with a point-by-point answer to the reviewers.

·

Basic reporting

The article #35559 shows two lactic acid bacteria identified as Weissella viridescens and Weisella confusa, which were isolated from fermented grapes with antagonistic activity against Salmonella Typhi and Salmonella Typhimurium. In addition, it was demonstrated that the cell-free supernatant of these lactic acid bacteria exerts the antagonistic activity and inhibit the formation of biofilms by blocking the signal of the AI-2, which is one of the three known types of autoinducers related to Quorum Sensing in Salmonella.
The manuscript is written correctly, clearly and unambiguously.
My first recommendation is about the title, when I read “Metabolites”, I think it is based on the identification of metabolites, so I consider authors could either perform new experiments for identifying the active metabolites or at least their chemical nature, or amend the title accordingly.
In the introduction section more details are required. In the second paragraph, it would be appropriate including information about the other types of autoinducers involved in Quorum Sensing in Salmonella (AI-1, AI-2 and AI-3) and, the reason for choosing AI-2. On the other hand, hypothesis is missing in this section, so it is suggested to include it based to the conclusion.
More than 70% of the references are from 2015 to 2019, which may be indicative of the current status of the addressed topic.

Experimental design

This research addresses a current topic within an adequate subject for publication in this journal.
I think the problem and research question are adequate.
The methods used are correct and in a general way are described correctly. In some cases the information may be enhanced:
1.- In Isolation of LAB from fermented foods method a reference is missing (lines 137-139) or mention it if you have validated that method previously.
2.- In Agar well diffusion test is important to mention the height the agar reaches in the plate or the volume of agar that was placed in each plate.
3.- In the MIC and MBC method (line 170), please specify if 105 CFU/mL is the final concentration or the concentration added in the 10 μl for each well and, also the final volume in each well must be indicated.
The developed experiments were designed correctly; however an experiment that helps to make a more specific identification would enrich the work.
The statistical analysis is accepted but the statistical design is missing. In the statistical analysis section, it is written “Turkey´s HSD test”, which could change for “Tukey´s HSD test”.

Validity of the findings

I wonder why authors mention the lactic acid bacteria isolated from fermented grapes are “potent” What do they mean with potent? It should be indicated based on which scale or respect to what?
Results show a good anti-salmonella and anti-biofilm activities of LAB-CFCS tested. Data presented seems robust, clear and sound, and conclusion is adequate.
In the discussion, in lines 409-410, it is mentioned that CFCS contain other metabolites than organic acids, which is not clear from the methodological approach or results obtained, so that these lines are speculative.

Additional comments

Abstract:
A period is missing after each subtitle, i.e., “Background. Salmonella Typhi…”

Introduction:
Citation in line 71, replace “Alum et al., 2016” by “Alum, Chukwu & Ahudie, 2016”.
Citation in line 72, is not correct, replace “Bhutia & Narain, 2010;…” by “Bhatia & Narain, 2010;…”

Materials & Methods:
In line 122, the word “this” could be changed for “the” or be omitted.
In line 214, the word “Salmonella” must be in italics.
In line 229, change the parenthesis position. “… as described by (Liu et al., 2014)…” “… as described by Liu et al. (2014)…”

Results:
In line 277, delete the letter “s” in “…Tables 2…”
In line 277, you mentioned “…19 selected LAB-CFCS…” but in the table 2, that number does not coincide.
In lines 323-324, the sentence “…Therefore, they were preliminarily identified as members of the genus Lactobacillus or Lactobacillus-like microorganisms (Liu et al.,2014).” must be change to Discussion and remove italics from “Liu”.

Discussion:
Please include in the manuscript discussion an explanation to the following questions:
Why does a bacterial strain have more antagonistic activity against Salmonella Typhi compared to Salmonella Typhimurium?
Why do two strain of the same bacteria exert different antagonistic activity against a bacterial pathogen?

References:
The reference in lines 529-531 is badly located; it should be in alphabetical order after “Rutherford ST, Bassler BL. 2012 ...”

Reviewer 2 ·

Basic reporting

Line 80 – reform the sentence of action driven salmonella having goals into more neutral
Line 81 – maybe to highlight the most common symptoms of food poisonings?
Line 85 – better wording for “bacteria sense them” or reforming wording?
Line 94 – Several serovars… move sentence before the previous sentence at Line 91 “Biofilm…” for more comprehensive and logical explanation order
Line 100 – A range … maybe to add also the food hygienic practices to emphasise the initial need to produce in hygienic and controlled conditions to AVOID the contamination possibilities as much as possible, especially in food production facilities.

Table 2 – Why are all the SDs 0.00 and all the values 10-folds only? Are these the correct values or table needs to be updated?

Tables and figures – clear, understandable and good amount of information

Introduction – clearly written in a good narrative order considering the relevant aspects of the subject in good detail and covering the wider aspects of the subject. Some minor polishing.

Experimental design

Line 134 – reduce word fermented to only the first for easier reading to avoid repetition

Materials and methods – well and clearly structured and written containing good amount of detailed information with appropriate references

Validity of the findings

Line 261-268 – replace this source of isolations with materials and methods line 134 description
Line 269 – Start new paragraph
Line 277 – Change Tables -> Table

Results – section is very well written, constructed and formatted to present the results in structured, clear and understandable way. The chain of validation of the functionality and effectiveness are well structured in the analysis to ensure best possible results with good outcomes.

Line 336 – This might be too straightforward conclusion of the usability and to compare with the more difficult options. Maybe this would be more justified if you can name some commonly used, or prominent control methods, which then are more difficult to apply in practice?

Line 367 – This due -> this is due
Line 412 – Remove -> It would not be surprising, -> These Weissella… OR Prior better understanding, these Weissella…

Additional comments

Very well and clearly written article with detailed information to support the experiments and findings. The study was well structured, and the presentation of the study and findings further highlight the authors’ abilities to present complex issues in clear format. The study is an interesting and topical, and the findings can be applied to wide range of applications and is very useful for further studies to be utilized.

I would want to encourage the authors to continue with this line of work and explore the possibilities to implement these findings in practice to ensure better food safety procedures in the future. This was a good study and hopefully the work will continue so that the findings will not remain only at the laboratory and academic level but are created into easy-to-use real-life options for industry.

---

## Round 0.2 · accepted · Accept

The manuscript has much improved and is now in suitable form for acceptance. Inclusion of details about the possible mechanisms of interference of biofilm formation is important as the paper contributes with practical aspects of potential biotechnological applications of LABs, as well as basic aspects of the molecular mechanisms involved.